# Key Genes of Lipid Metabolism and WNT-Signaling Are Downregulated in Subcutaneous Adipose Tissue with Moderate Weight Loss

**DOI:** 10.3390/nu11030639

**Published:** 2019-03-16

**Authors:** Ruth Schübel, Disorn Sookthai, Judith Greimel, Theron S. Johnson, Mirja E. Grafetstätter, Romy Kirsten, Mario Kratz, Cornelia M. Ulrich, Rudolf Kaaks, Tilman Kühn

**Affiliations:** 1German Cancer Research Center (DKFZ), Division of Cancer Epidemiology, Im Neuenheimer Feld 581, 69120 Heidelberg, Germany; ruth.schuebel@gmx.de (R.S.); disorn.s@gmail.com (D.S.); judithgreimel@gmail.com (J.G.); t.johnson@dkfz.de (T.S.J.); m.grafetstaetter@dkfz.de (M.E.G.); r.kaaks@dkfz.de (R.K.); 2Biobank of the National Center for Tumor Diseases (NCT) Heidelberg, Im Neuenheimer Feld 460, 69120 Heidelberg, Germany; romy.kirsten@nct-heidelberg.de; 3Institute of Pathology, Heidelberg University Hospital, Im Neuenheimer Feld 224, 69120 Heidelberg, Germany; 4Division of Public Health Sciences, Fred Hutchinson Cancer Research Center, Seattle, WA 98109, USA; mkratz@fredhutch.org; 5Huntsman Cancer Institute and Department of Population Health Sciences, University of Utah, 2000 Circle of Hope, Salt Lake City, UT 84112-5550, USA; neli@hci.utah.edu

**Keywords:** adipose tissue, gene expression, obesity, weight loss, transcriptomics

## Abstract

Smaller cross-sectional studies and bariatric surgery trials suggest that weight loss may change the expression of genes in adipose tissue that have been implicated in the development of metabolic diseases, but well-powered intervention trials are lacking. In post hoc analyses of data from a 12-week dietary intervention trial initially designed to compare metabolic effects of intermittent vs. continuous calorie restriction, we analyzed the effects of overall weight loss on the subcutaneous adipose tissue (SAT) transcriptome. Changes in the transcriptome were measured by microarray using SAT samples of 138 overweight or obese individuals (age range: 35–65 years, BMI range: 25–40, non-smokers, non-diabetics). Participants were grouped post hoc according to the degree of their weight loss by quartiles (average weight loss in quartiles 1 to 4: 0%, −3.2%, −5.9%, and −10.7%). Candidate genes showing differential expression with weight loss according to microarray analyses were validated by reverse transcription quantitative polymerase chain reaction (RT-qPCR), and fold changes (FCs) were calculated to quantify differences in gene expression. A comparison of individuals in the highest vs. the lowest weight loss quartile revealed 681 genes to be differentially expressed (corrected *p* < 0.05), with 40 showing FCs of at least 0.4. Out of these, expression changes in secreted frizzled-related protein 2 (SFRP2, FC = 0.65, *p* = 0.006), stearoyl-CoA desaturase (SCD, FC = −1.00, *p* < 0.001), and hypoxia inducible lipid droplet-associated (HILPDA, FC = −0.45, *p* = 0.001) with weight loss were confirmed by RT-qPCR. Dietary weight loss induces significant changes in the expression of genes implicated in lipid metabolism (SCD and HILPDA) and WNT-signaling (SFRP2) in SAT.

## 1. Introduction

Obesity, characterized by excessive accumulation of adipose tissue, is a risk factor for major chronic diseases such as type 2 diabetes, cardiovascular diseases, and many types of cancer [1,2,3]. Dysfunctional adipose tissue signaling is a hallmark of obesity and drives its comorbidities, e.g., via increased release of inflammatory factors, altered adipokine profiles, and constant anabolic stimuli by growth factors and hormones [3,4].

For the prevention of pathophysiological complications among overweight or obese people, it has been recommended to aim at 5–10% weight loss by calorie restriction (CR) approaches and/or increase in physical activity levels [5]. Although the central role of adipose tissue accumulation, particularly in the visceral compartment, is undisputed, there is limited knowledge on the reversibility of obesity-induced alterations in adipose tissue function with weight loss. Adipose tissue gene expression has been used to investigate obesity-related metabolic dysfunction in human trials, as it may provide the basis for the understanding of dysfunctional adipose tissue signaling and its reversibility by weight loss [6,7]. However, many previous studies have been cross-sectional studies with inter-individual comparisons of lean and obese individuals [8,9], or bariatric surgery interventions with substantial weight loss [10,11], while not much is known about the effects of diet-induced weight loss on the adipose tissue transcriptome from larger intervention trials. Moreover, only the expression of specific candidate genes was analyzed in many previous studies, while there is a lack of comprehensive transcriptome-wide, i.e., microarray-derived gene expression data from dietary intervention trials (see Appendix A for a systematic overview).

Here, we evaluated the effects of CR-induced weight loss on abdominal subcutaneous adipose tissue (SAT) gene expression in post hoc analyses from the HELENA Trial (Healthy nutrition and energy restriction as cancer prevention strategies: a randomized controlled intervention trial), a dietary intervention trial among 150 overweight and obese individuals. In this trial, participants achieved moderate weight loss of 7.1% (with intermittent calorie restriction in form of the 5:2 diet), 5.2% (continuous calorie restriction), and 3.3% (control group) over 12 weeks [12]. While the main analyses of the trial did not show differences in SAT gene expression across the three study arms, the goal of the present analyses was to evaluate whether the response to the dietary interventions, i.e., the overall degree of weight loss that was achieved by either method, had an impact on SAT gene expression. Furthermore, we analyzed if the protein products of genes differentially expressed with weight loss in SAT are similarly regulated in serum, and if gene expression and protein levels are related to body mass index (BMI) and established clinical markers of lipid and glucose metabolism (high density lipoprotein [HDL]-cholesterol, low density lipoprotein [LDL]-cholesterol, insulin, homeostatic model assessment for insulin resistance [HOMA-IR] values, and C-reactive protein [CRP]).

## 2. Materials and Methods

### 2.1. Study Design and Participants

One hundred fifty overweight or obese (BMI >25 and ≤40 kg/m²) men and women (age 35 to 65 years) participated in the HELENA Trial, a randomized controlled intervention study (Clinical Trials.gov, NCT02449148). The HELENA Trial was approved by the ethics committee of the medical faculty Heidelberg (University of Heidelberg, Germany) and had the aim to investigate the effect of timing of calorie restriction (intermittent vs. continuous) on metabolic outcomes including adipose tissue gene expression. Details on study design, power-calculation, recruitment procedures, study assessments, and intervention protocol have been published previously [12,13]. Briefly, all participants were non-smokers, did not have severe chronic diseases (diabetes, major cardiovascular diseases, cancer, or hepatic or kidney dysfunction) according to a medical interview and routine blood parameters prior to commencement of the trial, and did not take lipid-lowering medication. By block randomization, participants were assigned to one of the three study arms: intermittent calorie restriction (ICR; 5 days per week at 0% energy deficit and 2 days at 75% deficit), continuous calorie restriction (CCR; daily energy deficit 20%), or a control regimen (without energy limitations), after written informed consent had been obtained. All three groups received general recommendations for a healthy and balanced diet according to the official guidelines of the German Society for Nutrition [14]. At baseline and after the controlled intervention phase of 12 weeks, participants attended the study center for the study assessments, including anthropometric measurements (i.e., height, body weight, waist circumference, hip circumference), blood draw, and SAT biopsy. Body weight, body height, and blood pressure were assessed by trained personnel according to standard operating procedures [12,13]. Overall, six participants withdrew across the intervention phase. Two further participants had to be excluded from the second biopsy, one due to temporary use of anti-coagulants and the other one due to side effects (skin irritation) at the first biopsy (see Appendix A).

For the present project, the HELENA Trial cohort was categorized by quartiles of weight loss during the 12-week intervention phase, irrespective of the dietary method by which weight loss was achieved. These pooled analyses on the effects of overall weight loss on SAT gene expression were pre-specified in the study protocol [13] and were motivated by the fact that we had not observed significant differences between changes in gene expression by intermittent calorie restriction vs. continuous calorie restriction vs. control [12]. Since we aimed to investigate the effect of weight loss on SAT gene expression profiles, and because the number of participants with weight gain (*n* = 4 with weight gain of >2%) was too small for well-powered analyses, we decided to exclude these participates. Thus, a sample of 138 participants was used for the present analyses.

### 2.2. Blood-Based Biomarkers

Blood draws from peripheral veins at the arm were conducted after a minimum of eight hours of overnight fasting. Routine metabolic biomarkers (fasting glucose, HDL cholesterol, LDL cholesterol, and total cholesterol) were quantified at the Central Laboratory of the University Hospital Heidelberg immediately after the blood draw, while fresh blood samples were processed, aliquoted, and stored at −80 °C at the Biobank of the National Center for Tumor Diseases (NCT, Heidelberg, Germany) in accordance with the regulations of the Biobank.

Serum concentrations of C-reactive protein (CRP) and insulin were measured on the Quickplex SQ 120 instrument from Meso Scale Discovery (MSD, Rockville, MD, USA) by electrochemiluminescence (ECL) using the manufacturer’s proprietary kits. Hypoxia inducible lipid droplet-associated (HILPDA) and secreted frizzled-related protein 2 (SFRP2) were measured using enzyme-linked immunosorbent assays (ELISA; Emax Immunoassay System, Biozol, Eching, Germany). Intra-batch CVs were 9.1% and 3.6% for HILPDA and SFRP2. Both ECL and ELISA quantifications were performed at the Division of Cancer Epidemiology laboratories, German Cancer Research Center (DKFZ) Heidelberg, Germany. Repeat samples from individual participants were positioned on the same analytical batch for biomarker quantification.

### 2.3. Adipose Tissue Biopsies

Details on the procedures for the local abdominal SAT biopsies have been described previously [12,13]. Briefly, SAT samples were obtained under local anesthesia by needle aspiration approximately 10–12 cm lateral to the umbilicus. All participants completed a minimum of eight hours of overnight fasting prior to sample collections. The SAT samples were immediately rinsed with sterile saline, snap-frozen in liquid nitrogen, aliquoted, and stored at −80 °C at the Biobank of the National Center for Tumor Diseases (NCT, Heidelberg, Germany) for future analyses.

### 2.4. mRNA Extraction and Microarray Analyses

Total mRNA extraction from SAT was done using the RNeasy Plus Universal Mini Kit (QIAGEN, Hilden, Germany) run on the QIAcube^®^ (QIAGEN, Hilden, Germany), and the instructions of the manufacturer’s protocol were followed. Integrity of RNA samples was measured on a Bioanalyzer 2100 (Agilent Technologies, Palo Alto, CA, USA). Samples with an mRNA Integrity Number above seven were stored at −80 °C and used for microarray analyses with Human HT-12v4 Expression BeadChips (Illumina, San Diego, CA, USA), at the Genomics and Proteomics Core Facility of the German Cancer Research Center (DKFZ), Heidelberg.

The MicroArray data is available on ArrayExpress upon publication (accession number: E-MTAB-5926).

### 2.5. Reverse Transcription and Quantitative Polymerase Chain Reaction (RT-qPCR)

In accordance with the findings of our microarray experiment on differential gene expression by weight loss (see results section), three out of forty identified candidate genes (SCD, SFRP2, and HILPDA) were selected for RT-qPCR validation. This selection was made based on technical criteria, i.e., the feasibility to design primers, and scientific interest. For *SCD*, a central enzyme in fatty acid metabolism, it has previously been reported from smaller studies (see Appendix A) that adipose tissue gene expression was downregulated with weight loss, which we intended to verify. Moreover, SCD may play a role in the development of cancer [15], which motivated our selection. In addition, we selected *SFRP2* (implicated in WNT signaling, with a potential role in tumorigenesis [16]) and *HILPDA* (a potential regulator of lipid droplet formation and insulin sensitivity [17]) as more novel candidate genes. A validation of further genes was not possible due to the low amount of available fat tissue and due to financial restrictions.

Reverse transcription of mRNA was completed using SuperScript™ IV VILO™ (Invitrogen, ThermoFisher Scientific, Waltham, USA), while the total amount of mRNA was maintained across all reverse transcriptions. Validation of expression levels for selected candidate genes were performed by RT-qPCR (PikoReal™ 96 Real Time PCR System, ThermoScientific™, Waltham, MA, USA) using the SYBR Green Master Mix (DyNAmo ColorFlash SYBR Green qPCR Kit, ThermoScientific, Waltham, MA, USA). The RT-qPCR settings with three technical replicates were implemented in accordance to the MIQE Guidelines (Minimum Information for Publication of Quantitative Real-Time PCR Experiments) [18,19]. Specific oligonucleotides sequences for RT-qPCR validation of stearoyl-CoA desaturase (*SCD*), *SFRP2*, and *HILPDA* were designed using Primer-BLAST, while high performance liquid chromatography purified primers were purchased from Eurofins Genomics (Eurofins Scientific SE, Luxemburg). Sequence accession numbers and primer sequences were as follows: *HILPDA*: NM_013332.3, forward (5′→3′): TGTTAGGTGTGGTACTGACCC and reverse (5′→3′): CTCTGTGTTGGCTAGTTGGC; *SFRP2*: NM_003013, forward (5′→3′): ATGCTTGAGTGCGACCGTTT and reverse (5′→3′): TACCTTTGGAGCTTCCTCGG; *SCD*: NM_005063.4, forward (5′→3′): TCCAGAGGAGGTACTACAAACCT and reverse (5′→3′): GCACCACAGCATATCGCAAG. Expression levels of the target genes were normalized to the reference gene *IPO8* (NM_006390.2, forward (5′→3′): CGAGCTCAACCAGTCCTACA and reverse (5′→3′): TCTGGCCAGTATTGTGTCACC) and analyzed with the delta-delta Ct method [20]. The criteria for selection of the reference gene were that IPO8 was not involved in energy metabolism, the expression level of *IPO8* was above the lower limit of detection in our microarray experiment, and the microarray data showed no changes in the expression levels between baseline and Week 12. Overall, specificity of the primers was insured by melt-cure analysis and 2D gel-electrophoresis. For two participants, mRNA sample volumes were limited so that the microarray gene expression results could not be validated by RT-qPCR.

### 2.6. Literature Overview

To provide an overview of previous studies on intentional weight loss induced by dietary interventions or bariatric-surgery and changes in the adipose tissue transcriptome (assessed by microarray analyses), we conducted a systematic literature search in PubMed. The identified studies are summarized in Appendix A.

### 2.7. Statistical Analyses

Pre-processing of microarray data (log2 transformation, imputation of missings [21], and batch standardization by ComBat [22]) was conducted in Chipster 3.8. The analyses on gene expression profiles per weight loss categories were obtained on the basis of linear models with age and sex adjustment, using the limma package in R (R Foundation for Statistical Computing) including Benjamini Hochberg correction for multiple testing. We did not further adjust for the initial study arm (intermittent calorie restriction vs. continuous calorie restriction vs. control), as the study arms were not associated with differences in SAT gene expression [12], and statistical adjustment thus only very marginally affected the statistical estimates. The same was true regarding BMI at baseline. Pathway analyses on weight loss-associated gene expression changes were done with parametric analyses of gene-set enrichment (PAGE), using the piano package in R including all significantly (*p* < 0.05) differentially regulated genes between lowest and highest weight loss quartile.

We conducted linear regression models in SAS 9.4 (Cary, NC, USA), adjusted for age and sex to analyze the overall effect of weight loss on log2 fold changes in both microarray- and RT-qPCR-derived expression levels of SFRP2, HILPDA, and SCD. In this context, p-values for pairwise comparison of least square means in gene expression between quartiles of weight loss (with the lowest weight loss quartile as the reference) were based on a t-test.

Linear mixed models for repeated measurement with age and sex adjustment were carried out in SAS 9.4 (Cary, NC, USA) to analyze time by treatment interaction effects on serum HILPDA and SFRP2 levels. The data on concentration levels are shown as mean values ± SD (with normal 95% CIs) and relative changes over time as mean ± SEM of log percentage changes.

Cross-sectional correlations between gene expression levels, BMI, and circulating biomarkers (HDL, LDL, insulin, HOMA-IR, CRP, SFRP2, HILPDA) were assessed by Spearman`s coefficients adjusted for age and sex in R.

## 3. Results

### 3.1. Characteristics of the Study Cohort

The study group (*n* = 138) was categorized by weight loss as follows: Quartile 1 (*n* = 34; ≤2% weight gain to ≤1.9% weight loss, average: 0%), Quartile 2 (*n* = 35; weight loss from >1.9% to ≤4.5%, average: 3.2%), Quartile 3 (*n* = 34, weight loss from >4.5% to ≤7.5%, average: 5.9%), and Quartile 4 (*n* = 35, weight loss >7.5%, average: 10.7%); participants in the four quartiles had similar baseline characteristics with regard to, age, sex, BMI, and routine metabolic biomarkers (see Table 1).

### 3.2. Microarray Gene Expression by Weight Loss Groups

There were 681 differentially expressed genes across weight loss groups (*p*_adjusted_ < 0.05). Out of these, 40 genes showed an absolute log_2_ FC of at least 0.4 between the lowest and the highest weight loss quartile (see Table 2). Fatty acid desaturase 1 (*FADS1*, FC = −1.09, *p*_adjusted_ < 0.001), *SCD* (FC = −1.00, *p*_adjusted_ < 0.001), glycerol-3-phosphate acyltransferase (*GPAM*, FC = −0.54, *p*_adjusted_ < 0.001), and diazepam binding inhibitor (*DBI)* (FC = −0.44, *p*_adjusted_ < 0.001), i.e., genes encoding key enzymes of lipid synthesis and cholesterol/fatty acid transport, were significantly downregulated among participants in the highest weight loss quartile. Likewise, several genes with functions in extracellular matrix modeling, cell–cell signaling, or cell differentiation were significantly downregulated (e.g., collagen, type XV, alpha 1 (*COL15A1*, FC = −0.50, *p*_adjusted_ < 0.001), *SFRP2* (FC = −0.65. *p*_adjusted_ = 0.006), and *HILPDA* (FC = −0.45, *p*_adjusted_ = 0.001)). Highest fold changes among upregulated genes with weight loss were observed for complement component 6 (*C6*, FC = 0.66, *p*_adjusted_ = 0.001) and cell death-inducing DFFA-like effector A (*CIDEA*, FC = 0.47, *p*_adjusted_ = 0.015).

Parametric analyses of gene-set enrichment (PAGE), performed on microarray data for all 681 significantly differentially regulated genes (*p*_adjusted_ < 0.05), revealed a downregulation of lipid metabolism, cell membrane, and oxidation processes (e.g., GO:0006633_fatty acid biosynthetic process, GO:0035338_long-chain fatty-acyl-CoA biosynthetic process and GO:0006631_fatty acid metabolic process, GO:0005886_plasma membrane, and GO:0016491_oxidoreductase activity) (see Figure 1 and Appendix A for the top 50 enriched pathways).

### 3.3. Validation of Gene Expression Levels for SCD, SFRP2, and HILPDA

Consistent with the strong results for *SCD* in our microarray experiment, RT-qPCR validation revealed significant differences in *SCD* expression across weight loss groups (*p* < 0.001; see Figure 2). The comparison of microarray and RT-qPCR data for *SFRP2* and *HILPDA* shown in Figure 2 illustrates the overlapping results for differential expression with increasing weight loss with both quantification methods. Downregulation of these genes across weight loss groups followed a dose–response gradient.

### 3.4. Circulating Levels of HILPDA and SFRP2

Based on the results for downregulation of *HILPDA* and *SFRP2* genes with weight loss, we selected these novel candidates to investigate if there are changes in circulating levels of the encoded proteins, again by weight loss quartiles. There were no significant differences (*p* > 0.05) between the weight loss quartiles for changes in circulating HILPDA or SFRP2 levels (see Table 3). Correlations between adipose tissue gene expression levels and corresponding serum protein concentrations of HILPDA and SFRP2 were low (ρ < 0.2) and not significant, both in cross-sectional analyses and with regard to changes in levels over time (see Figure 3 and Appendix A).

### 3.5. Correlation of Adipose-Tissue Gene Expression with BMI and Metabolic Biomarkers

Analyses on the correlates (BMI, HDL, LDL, insulin, HOMA-IR, and CRP) of gene expression levels of *SFRP2*, *HILPDA*, and *SCD* revealed a modest positive correlation between *SFRP2* and BMI (ρ = 0.42, see Figure 3). *HILPDA* and *SCD* showed no meaningful correlations (ρ > 0.4) with BMI or routine metabolic markers. There were no correlations between circulating SFRP2 and HILPDA levels and BMI or other metabolic markers. Results on cross-sectional analyses at Week 12 (Figure 3) were highly similar to those from baseline (Appendix A). Analyses on changes in gene expression and biomarker levels over time showed only weak correlations at ρ < 0.4 (Appendix A). Correlations between expression levels of all 40 genes differentially regulated with weight loss in adipose tissue and BMI as well as established circulating markers of metabolism are shown in Appendix A. Overall, correlations with BMI and classical disease risk biomarkers were weak.

## 4. Discussion

In the present study we found that moderate weight loss induces differential expression of key enzymes of lipid synthesis and cholesterol transport as well as extracellular matrix remodeling in SAT. *SCD*, *SFRP2*, and *HILPDA* were identified and validated as novel adipose tissue biomarkers related to weight change. However, we observed no correlations between SAT gene expression levels of *SFRP2* and *HILPDA* with blood levels of the proteins. Neither weight loss (prospectively) nor BMI (cross-sectionally) were related to circulating SFRP2 and HILPDA, suggesting that these adipose changes may not affect systemic blood levels. BMI showed a modest correlation with the expression of *SFRP2*, but not with the expression of *SCD* and *HILPDA*.

We observed pronounced effects of weight loss on the expression of *SFRP2*, a key modulator of WNT-signaling, in adipose tissue. Known functionality of SFRPs are modulations of growth factors belonging to the wingless-type mouse mammary tumor virus integration site (WNT) family. In general, WNT-signaling regulates cell differentiation and tissue remodeling in several tissue types including adipose tissue [23]. SFRP1, −2 and −4 bind to WNT-molecules and inhibit their actions with the consequence that adipogenesis is induced [24,25,26]. Against this background it is interesting to note that *SFRP2* and −*4* are significantly downregulated in SAT among gastrointestinal cancer patients with cachexia compared to weight stable patients [27]. Our finding on reduced *SFRP2* expression with intentional weight loss supports the microarray-based result from two previous studies with approximately 10% weight loss, induced by low calorie diets [28,29]. Similarly, a significantly higher expression of *SFRP2* and −*3* in human SAT and VAT samples from obese individuals compared to non-obese has been reported, with more pronounced differences in VAT [30]. Expression levels for *SFRP2* were positively correlated with insulin sensitivity (HOMA-IR; r = 0.49) and BMI (r = 0.65) in that study [30], while slightly weaker correlations were observed in the present project (HOMA-IR r = 0.28 and BMI r = 0.42). We did not observe a correlation for *SFRP2* gene expression levels in SAT and circulating SFRP2. Moreover, unlike in one previous study [31] that showed significant positive associations between circulating SFRP2 and BMI as well as insulin levels, there were no such correlations in our study. Nevertheless, our data suggest that *SFRP2* expression in adipose tissue can be controlled by moderate intentional weight loss. The implication of modifications in adipose tissue *SFRP2* expression for chronic diseases such as cancer [16,32] requires further studies, however.

*HILPDA* is upregulated in both the VAT and SAT fraction of morbidly obese patients receiving bariatric surgery [17], which is in line with the present finding of decreased *HILPDA* expression in SAT with intentional CR-induced weight loss. There is one previous report indicating the same direction of change in *HILPDA* expression levels following a low-calorie diet for weight loss, although the FC magnitude was lower [29]. The highly conserved protein encoded by *HILPDA* is associated with lipid droplet formation especially in hepatocytes and adipocytes [33]. As a target of peroxisome proliferator activated receptor alpha (PPARα), HILPDA might be involved in hepatic triglyceride turnover and be linked to insulin sensitivity [34]. Initially, HILPDA was identified in cervical cancer cells with activation under hypoxic conditions [35], which is why it was first termed hypoxia induced protein 2 (HIG2). Our results are particularly intriguing, because the functional involvement of HILPDA in cancer-related WNT-signaling and the potential use of this molecule as a novel therapeutic target for cancer treatment is under discussion [36,37]. 

The strongest changes in SAT gene expression with CR-induced weight loss in our study were observed for *SCD* and *FADS1*, both rate limiting enzymes in the biosynthesis of unsaturated fatty acids. These results are in agreement with previous reports from CR intervention and bariatric surgery trials [6,7,28,29,38,39]. Heightened activity of SCD, which catalyzes the synthesis of monounsaturated fatty acids, has been postulated to be a major checkpoint in the pathogenesis of obesity-driven diseases, in particular type 2 diabetes, cardiovascular diseases, and cancer [40,41]. *FADS1* and *FADS2* encode the proteins Δ5-desaturase and Δ6-desaturase which catalyze the conversion of linoleic acid and α-linolenic acid into longer chained polyunsaturated fatty acids with more double bonds [42]. Large-scale prospective studies revealed that the activity levels of these enzymes are associated with the risk for type 2 diabetes, with the activity of Δ5-desaturase being inversely and the activity of Δ6-desaturase being directly related to disease risk [42]. In addition, there is evidence to suggest that altered FADS1 activity is at the interface between obesity and other major chronic diseases [43]. Future research is required to further unravel the metabolic consequences of decreased *FADS1* gene expression in adipose tissue with weight loss.

There were several additional genes among our top hits that confirm findings from previous transcriptomics analyses of adipose tissue, which we identified by a systematic literature search on microarray-based trials for differential adipose tissue gene expression after CR- and bariatric surgery-induced weight loss (overlaps to our study are shown in Appendix A). Interestingly, a recently published analysis from the DIOGENES trial including 191 obese participants undergoing a low-calorie diet for 8 weeks with ~10% weight loss, revealed 350 genes that were differentially regulated after weight loss (absolute FC ≥ 0.2 and corrected *p*-values < 0.05) [29]. In this study, 37 genes showed FCs with weight loss at a magnitude ≥0.4 (the fold change cut-point in our study) [29], out of which *SCD*, *SFRP2*, *THBS4*, and *GPAM* were among the top hits in our study as well (i.e., FC ≥ 0.4) [29].

The main strength of this study was the size of the study cohort, which was entirely characterized by microarray-based gene expression profiling, with additional validation of the top hits by RT-qPCR. Nevertheless, it must be acknowledged that we could not use VAT samples for gene expression analyses, as taking VAT biopsies was not possible for ethical and practical reasons. It was neither feasible to separate the tissue biopsy into different cellular fractions prior to mRNA extraction. It also should be noted that our trial comprised a rather homogeneous group of overweight and obese individuals without metabolic co-morbidities. Thus, we cannot generalize our results to other study populations (e.g., diabetics, morbidly obese individuals, or individuals undergoing stronger weight fluctuations). Unlike previous studies, in which SAT gene expression experiments facilitated the identification of circulating adipokines [44], we did not observe associations between SAT gene expression levels and blood levels of SFRP2 and HILPDA. We did not have the opportunity to expand our work to laboratory-based mechanistic analyses, and functional aspects of our findings from post hoc analyses of RCT data need to be investigated by controlled wet lab studies before firm conclusions on biological mechanisms can be drawn. Finally, the number of genes that we could validate was restricted due to financial reasons and limited adipose tissue sample availability.

## 5. Conclusions

In summary, our study showed differential expression of genes in subcutaneous adipose tissue encoding proteins, which may reduce lipid biosynthesis, facilitate lipid utilization, alter WNT-signaling, and convert adipose tissue structure with weight loss. Mechanistic studies are needed to investigate whether proteins such as SFRP2, HILPDA, and SCD mediate associations between obesity and its comorbidities.

## Figures and Tables

**Figure 1 nutrients-11-00639-f001:**
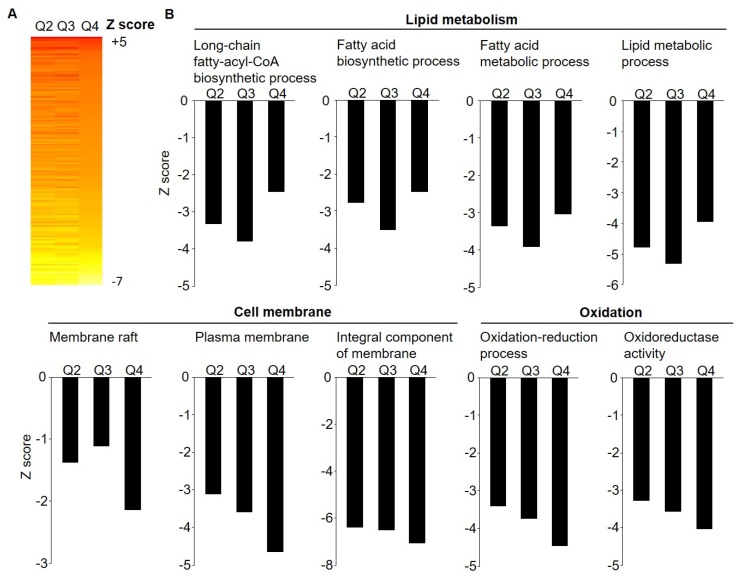
Effect of weight loss on SAT gene expression pathway enrichment (*n* = 138). Parametric analysis of gene-set enrichment (PAGE) were performed on microarray data including 681 significantly (*p* < 0.05) regulated genes across weight loss quartiles. (**A**) Heat map on biological pathways that were significantly differentially regulated by weight loss, based on the Z score for difference between weight loss quartiles with Q1 (lowest quartile) as a reference. (**B**) Biological pathways involved in lipid metabolism, cell membrane, and oxidation processes were significantly downregulated with weight loss. Further list of pathways see Appendix A.

**Figure 2 nutrients-11-00639-f002:**
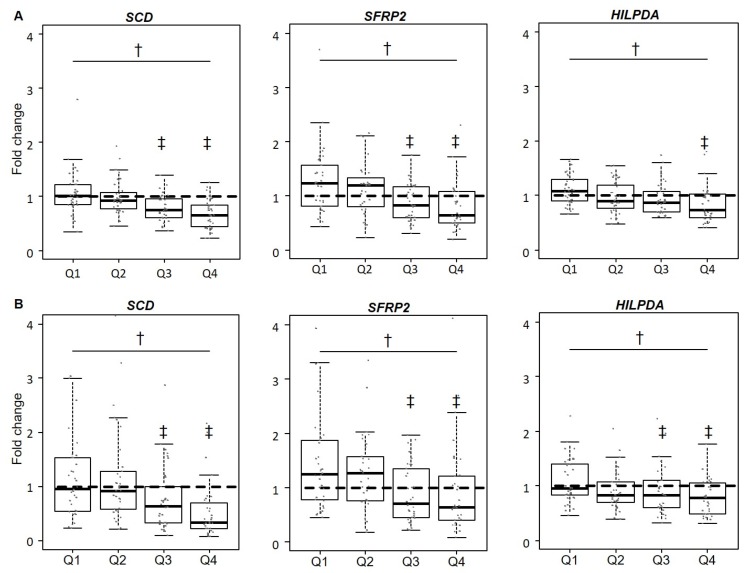
Effect of weight loss on SAT mRNA gene expression (microarray and RT-qPCR) of stearoyl-CoA desaturase (*SCD*), secreted frizzled-related protein 2 (*SFRP2*), and hypoxia inducible lipid droplet-associated protein (*HILPDA*). (**A**) Relative gene expression across weight loss quartiles (Q1: lowest quartile) determined by microarray (*n* = 138) is expressed as fold change (FC) between baseline (T_0_) and after 12 weeks (T_1_) (FC = 2^(logT1−logT0)^). (**B**) Expression profiles quantified by RT-qPCR (*n* = 136) are presented as FC of the delta-delta CT method (2^−ΔΔCT^). Data points with an FC > 4.2 are not plotted (*SCD*: *n* = 2, *SFRP2*: *n* = 1). Significant differences (*p* < 0.05) are depicted by † for overall effects and ‡ for pairwise comparisons.

**Figure 3 nutrients-11-00639-f003:**
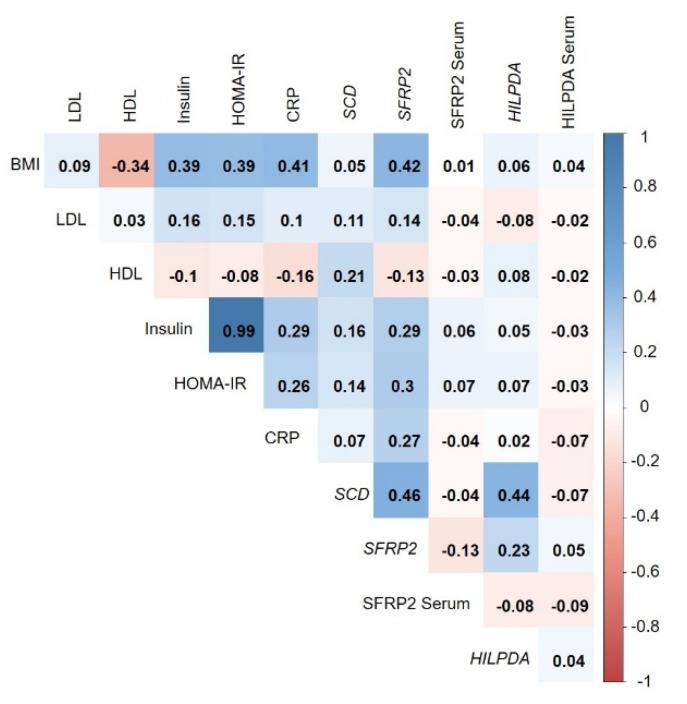
Cross-sectional correlations (at Week 12) between *SCD*, *SFRP2*, and *HILPDA* microarray gene expression levels, serum levels of SFRP2 and HILPDA, BMI, and metabolic biomarkers (LDL, HDL, insulin, HOMA-IR, CRP). Data are shown as Spearman coefficients adjusted for age and sex. The cross-sectional correlations for baseline data and for log relative changes between baseline and Week 12 are shown in Appendix A.

**Table 1 nutrients-11-00639-t001:** Baseline characteristics of the study cohort by weight loss quartiles (138) ^1^.

	Quartile 1	Quartile 2	Quartile 3	Quartile 4
	(*n* = 35)	(*n* = 34)	(*n* = 35)	(n = 34)
Women, *n* (%)	17 (48.6)	18 (52.9)	14 (40.0)	18 (52.9)
Age (years)	47.2 ± 8.3	52.0 ± 7.4	50.6± 8.7	50.6 ± 6.5
Weight (kg)	94.1 ± 14.4	93.2 ± 15.9	94.4 ± 13.7	94.5 ± 16.3
Height (cm)	173.4 ± 8.0	173.7 ± 10.9	173.6 ± 9.4	172.5 ± 10.3
BMI (kg/m^2^)	31.5 ± 3.7	30.8 ± 3.4	31.3 ± 3.8	31.7 ± 4.2
Education level, *n* (%) ^2^				
Primary school	4 (12.1)	2 (5.9)	5 (14.7)	2 (5.7)
Secondary school	14 (42.4)	9 (26.4)	6 (17.7)	8 (22.9)
Higher education	15 (45.5)	23 (67.7)	23 (67.6)	25 (71.4)
Glucose (mg/dL)	91.5 ± 7.9	95.1 ± 6.6	93.0 ± 6.6	93.0± 7.8
Insulin (mU/L)	11.0 ± 5.3	10.6 ± 5.0	13.0 ± 7.4	13.6 ± 7.0
HOMA-IR	2.5 ± 1.3	2.5 ± 1.2	3.0 ± 1.8	3.1 ± 1.6
Cholesterol (mg/dL)	212.8 ± 35.2	199.7 ± 31.7	214.9 ± 36.8	202.9 ± 34.9
HDL-cholesterol (mg/dL)	53.8 ± 15.9	51.8 ± 13.5	57.6 ± 13.7	52.9 ± 15.1
LDL-cholesterol (mg/dL)	128.8 ± 24.8	120.8 ± 25.1	129.2 ± 26.8	128.4 ± 29.8

^1^ Data are shown as means ± SD. ^2^ Education level is missing for two participants.

**Table 2 nutrients-11-00639-t002:** Up- and downregulated genes (*p*-value < 0.05; log_2_ fold change >0.4) for comparisons between highest- and lowest weight loss quartile (*n* = 138).

Address	Gene	Description	Highest vs. Lowest		
ID	Symbol		Weight Loss Quartile		
			Log_2_ FC ^1^	*p* Value ^1^	*p*_trend_ Value ^2^
**Downregulated genes**					
2360020	FADS1	Fatty acid desaturase 1	−1.09	<0.001	<0.001
4850195	SCD	Stearoyl-CoA desaturase	−1.00	<0.001	<0.001
2140128	SCD	Stearoyl-CoA desaturase	−0.69	<0.001	<0.001
3060639	SFRP2	Secreted frizzled-related protein 2	−0.65	0.006	0.004
6200253	THBS4	Thrombospondin 4	−0.58	0.019	0.026
7040372	GPAM	Glycerol-3-phosphate acyltransferase	−0.54	<0.001	0.001
7330544	ALDOC	Aldolase C, fructose-bisphosphate	−0.52	<0.001	0.001
3840026	GPAM	Glycerol-3-phosphate acyltransferase	−0.51	<0.001	<0.001
5090026	COL15A1	Collagen, type XV, alpha 1	−0.50	<0.001	<0.001
5550292	KLB	Klotho beta	−0.49	<0.001	<0.001
240400	PMEPA1	Prostate transmembrane protein	−0.48	0.002	0.001
4010709	NNAT	Neuronatin	−0.47	0.008	0.005
3390326	ME1	Malic enzyme 1, NADP(+)-dependent	−0.47	<0.001	<0.001
5550379	CAV1	Caveolin 1	−0.47	0.0041	0.017
7650053	ECHDC1	Enoyl CoA hydratase domain containing 1	−0.46	<0.001	<0.001
450292	TUBB2A	Tubulin	−0.46	0.008	0.011
7320441	HILPDA	Hypoxia inducible lipid droplet-associated	−0.45	<0.001	<0.001
4480341	DHCR24	24-dehydrocholesterol reductase	−0.45	<0.001	<0.001
2370041	LRRN3	Leucine rich repeat neuronal 3	−0.45	<0.001	0.0012
3450537	DGAT2	Diacylglycerol O-acyltransferase 2	−0.44	0.015	0.016
7610128	KANK4	KN motif and ankyrin repeat domains 4	−0.44	0.032	0.012
2480338	DBI	Diazepam binding inhibitor	−0.44	<0.001	<0.001
730040	LAMB3	Laminin, beta 3	−0.43	<0.001	<0.001
3830041	PMEPA1	Prostate transmembrane protein	−0.43	0.0061	0.0043
360192	INSIG1	Insulin induced gene 1	−0.43	<0.001	<0.001
3180048	IDH1	Isocitrate dehydrogenase 1	−0.42	<0.001	<0.001
2230538	LRRN3	Leucine rich repeat neuronal 3	−0.41	0.002	0.003
1710484	FMOD	Fibromodulin	−0.41	0.002	0.002
3390343	SREBF1	Sterol regulatory element binding	−0.41	0.029	0.022
3170594	MAL2	Mal, T-cell differentiation protein 2	−0.41	0.013	0.020
5310634	FASN	Fatty acid synthase	−0.41	0.013	0.0118
520474	TENM4	Teneurin transmembrane protein 4	−0.40	<0.001	<0.001
**Upregulated genes**					
6520040	C6	Complement component 6	0.66	<0.001	<0.001
6280370	LOC646688	Predicted: misc_RNA	0.51	<0.001	<0.001
2140278	CIDEA	Cell death-inducing DFFA-like effector a	0.47	0.015	<0.001
2490612	ADH1B	Alcohol dehydrogenase IB	0.45	<0.001	<0.001
7560543	MOCS1	Molybdenum cofactor synthesis 1	0.44	<0.001	<0.001
10048	CIDEA	Cell death-inducing DFFA-like effector a	0.44	0.012	<0.001
5490019	GPX3	Glutathione peroxidase 3	0.41	0.003	<0.001
6270372	EGFLAM	EGF-like, fibronectin type III and laminin G domains	0.40	0.012	<0.001

^1^*p* value (adjusted with Benjamini Hochberg) and log_2_ fold change (FC) calculations were based on a linear model from the *limma* package in *R* for highest versus lowest weight loss quartile at Week 12. ^2^
*p*_trend_ value (adjusted with Benjamini Hochberg) was based on linear model with weight as continuous covariate.

**Table 3 nutrients-11-00639-t003:** Change in serum concentrations of HILPDA and SFRP2 by weight loss quartiles ^1^.

	*n*	Baseline		*n*	Week 12		Relative	*p*-Value ^1^	
		Mean ± SD	95% CI		Mean ± SD	95% CI	Change	All	Q1 vs. Q4
HILPDA (µg/mL)									
Quartile 1	33	51.8± 26.5	(44.0, 59.6)	32	52.8 ± 24.2	(45.5, 60.1)	6.8 ± 4.8	0.78	0.84
Quartile 2	34	55.6 ± 25.3	(48.2, 62.9)	33	53.8 ± 23.9	(46.8, 60.9)	0.5 ± 7.4		
Quartile 3	32	49.0 ± 28.5	(40.5, 57.6)	34	48.9 ± 27.4	(41.0, 56.9)	4.3 ± 6.3		
Quartile 4	35	53.2 ± 29.0	(44.9, 61.5)	34	54.4 ± 28.5	(46.2, 62.7)	5.8 ± 9.2		
SFRP2 (ng/mL)									
Quartile 1	32	36.5 ± 29.5	(27.6, 45.3)	27	38.1 ± 34.1	(26.9, 49.3)	2.8 ± 7.1	0.36	0.31
Quartile 2	31	45.6 ± 26.5	(37.5, 53.7)	30	39.3 ± 19.5	(33.3, 45.4)	−3.0 ± 5.3		
Quartile 3	31	29.3 ± 13.6	(25.2, 33.4)	32	33.2 ± 15.3	(28.7, 37.8)	11.2 ± 8.9		
Quartile 4	33	45.0 ± 27.5	(36.9, 53.1)	31	46.1 ± 26.1	(38.2, 54.0)	−3.6 ± 8.1		

^1^*p* values for time by treatment interactions were calculated with linear mixed models adjusted for age and sex. Abbreviations: HILPDA, hypoxia inducible lipid droplet-associated protein; SFRP2, secreted frizzled-related protein 2.

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
