# Peer review of "Key Genes of Lipid Metabolism and WNT-Signaling Are Downregulated in Subcutaneous Adipose Tissue with Moderate Weight Loss"

_nutrients, 2019, doi:10.3390/nu11030639_

Round 1
Reviewer 1 Report
After modification according to reviewers comments, this reviewer believe the revised version of nutrients-432689 is much more scientifically sound and well described.
Reviewer 2 Report
none.
This manuscript is a resubmission of an earlier submission. The following is a list of the peer review reports and author responses from that submission.
Round 1
Reviewer 1 Report
Authors aimed to study the effects of 12-wks of continuous and intermittent caloric restriction in overweight and obese individuals (female and male) on adipose tissue lipid metabolism markers. However, some concerns limit the convincingness of this study. General and specific comments are described as below:
General comments:
-Abstract is confusing. Please revise methods and results section. It’s not clear what you studied and what is new in your study.
- How do you formulate the following goal: “The goal was to investigate the effects of moderate weight loss on adipose tissue transcriptional control.”? How do you know that the weight loss is moderate?
Please revise the aim of the study. Is not clear for the reader.
- Introdution: please do a better job revising the state of art and please justify why you perform this study?
- Why did you select stearoyl-CoA desaturase (SCD), hypoxia inducible lipid droplet-associated (HILPDA) secreted frizzled-30 related protein 2 (SFRP2) and so on…. Please justify this selection based on literature.
- Based on what you say that “all individuals did not have severe chronic diseases, like diabetes, major cardiovascular diseases? Did you performed any analysis to have sure of this ?
Specific comments:
Line 59. Please include abbreviation of HELENA.
Line 63. BMI abbreviation is missing.
HOMA., LDL HDL abbreviations are missing
Author's reply:
Reviewer 1
Authors aimed to study the effects of 12-wks of continuous and intermittent caloric restriction in overweight and obese individuals
(female and male) on adipose tissue lipid metabolism markers. However, some concerns limit the convincingness of this study.
General and specific comments are described as below:
General comments:
1. Abstract is confusing. Please revise methods and results section. It’s not clear what you studied and what is new in your study.
We have re-written parts of our abstract and hope that it is clearer now.
See lines 21 to 37
2. How do you formulate the following goal: “The goal was to investigate the effects of moderate weight loss on adipose tissue
transcriptional control.”? How do you know that the weight loss is moderate?
We know about weight loss among our study participants from physical examinations at baseline and after the 12-week
intervention, which we have stated in the methods section in more detail now.
We agree with the reviewer that the definition of “moderate” is subjective, although we think that the weight loss achieved in our
study (10.7 % in the highest weight loss quartile) is moderate compared to what can be achieved with very low calorie diets or
bariatric surgery. We have deleted the word “moderate” now, and have specified the average weight loss in our abstract to avoid
confusion. We have also rewritten the last part of our introduction for more clarity.
See lines 68 and 73 and lines 98 to 99
3. Please revise the aim of the study. Is not clear for the reader.We have added a more explicit statement about the aim of our study, as suggested by the reviewer.
See lines 66 to 73
4. Introduction: please do a better job revising the state of art and please justify why you perform this study?
The text on the aims of our study has been revised (see point 3).We thought that a systematic review of the literature is overly long for an introduction, but have covered the state-of-the-art, i.e. the evidence from previous similar studies, in detail in our
supplementary appendix, which we refer to now. We have also added a more explicit justification of our analyses.
See lines 58 to 60 and lines 68 to 73
5. Why did you select stearoyl-CoA desaturase (SCD), hypoxia inducible lipid droplet-associated (HILPDA) secreted frizzled-30
related protein 2 (SFRP2) and so on…. Please justify this selection based on literature.
As stated in our limitations section, we did not have the opportunity to quantify the expression of more than three genes due to limited sample availability and financial restrictions. We have made this more transparent in the methods section now, too.
The selection of three out forty genes that were identified by microarray for qPCR validation was based on literature review, as
stated in the methods section, even though it is possible that other researchers may have prioritized other candidate genes according to personal preferences. We have added the references that made us think SCD, HILDPA, and SFRP2 were of special relevance to the methods section.
See lines 147 to 157
6. Based on what you say that “all individuals did not have severe chronic diseases, like diabetes, major cardiovascular diseases? Did
you performed any analysis to have sure of this?
We can rule out that individuals with major chronic diseases entered our study, as we carried out detailed medical screening
including a comprehensive clinical biochemistry assessment before the study started. We have stated this in the methods section now.
See lines 90 to 91
7. Specific comments:
Line 59. Please include abbreviation of HELENA.
Line 63. BMI abbreviation is missing.
HOMA., LDL HDL abbreviations are missing
We have added the full terms and the full study title.
See lines 67 to 67 and 76 to 78
Reviewer 2 Report
The authors investigated the effects of weight loss on subcutaneous adipose tissue transcriptome in overweight or obese individuals. They showed that, compared to the lowest weight loss group, key genes of lipid metabolism (SCD and HILPDA) and WNT-signaling (SERP2) were downregulated in the highest weight loss group. This study is potentially interesting, but has some problems with the study design and results.
- The population in this study consists of intermittent calorie restriction, continuous calorie restriction and control groups. In their previous HELENA trial, the intermittent calorie restriction tended to decrease body weight compared to the continuous calorie restriction (p=0.053) and control (p<0.01). Although the authors mentioned that significant differences of gene expression had not been observed between the three groups, the effects of intervention groups would not be eliminated completely. The results of statistical analyses should be also adjusted by the intervention groups.
- The rationale why participants with more than 2% weight gain were excluded from this study population is unclear. Should the participants with weight gain be excluded, if the authors aim to investigate the effect of weight loss on adipose tissue gene expression profiles?
- The results in this study are somewhat deliberate. The rationale why SCD, HILPDA and SERP2 were selected is unclear. Also, why the authors do not investigate up-regulated genes by weight loss?
- As the authors mentioned in Discussion section, visceral adipose tissue is more relevant to the physiology of weight loss than subcutaneous adipose tissue. The authors should describe “subcutaneous adipose tissue” in their title.
Authors' Reply to Reviewer 2:
Reviewer 2
The authors investigated the effects of weight loss on subcutaneous adipose tissue transcriptome in overweight or obese individuals. They
showed that, compared to the lowest weight loss group, key genes of lipid metabolism (SCD and HILPDA) and WNT-signaling (SERP2) were downregulated in the highest weight loss group. This study is potentially interesting, but has some problems with the study design
and results.
1. The population in this study consists of intermittent calorie restriction, continuous calorie restriction and control groups. In their previous HELENA trial, the intermittent calorie restriction tended to decrease body weight compared to the continuous calorie restriction (p=0.053) and control (p<0.01). Although the authors mentioned that significant differences of gene expression had not been observed between the three groups, the effects of intervention groups would not be eliminated completely. The results of statistical analyses should be also adjusted by the intervention groups.
The reviewer raises a valid point. In fact, individuals in the intermittent and continuous calorie restriction groups were more
likely to be in the highest weight loss quartile than individuals in the control group. That said, an adjustment for study arm affected the observed associations only very marginally. This is related to the fact that values for potential confounders as well as baseline expression levels across the weight loss quartiles, which we generated in a post hoc manner, were very similar. We also tested whether initial BMI had an effect on the statistical associations, but estimates remained almost unchanged when adjusting for BMI. We have clarified this in the text now.
See lines 193 to 196
2. The rationale why participants with more than 2% weight gain were excluded from this study population is unclear. Should the
participants with weight gain be excluded, if the authors aim to investigate the effect of weight loss on adipose tissue gene expression profiles?
In general we agree with the reviewer that it would have been very interesting to have a group with weight gain in the study. However, there were too few weight gainers in our trial (n=4) to facilitate wellpowered analyses. As our goal was to analyze effects of weight loss, we found it hard to justify including these weight gainers. However, it should be noted than an inclusion of the weight gainers hardly affected our results (the reported signficant associations remained the same, likely due to the small number of weight gainers). We have commented on our exclusion in more detail in the methods section.
See lines 109 to 112
3. The results in this study are somewhat deliberate. The rationale why SCD, HILPDA and SERP2 were selected is unclear. Also, why
the authors do not investigate up-regulated genes by weight loss?
The reviewer is right that a validation of more genes would have been of great interest. As stated in our responses to Reviewer 1
above, we only had the chance to validate a limited number of genes, and the selection of genes for validation remains somewhat
subjective (despite our comprehensive literature review). We motivate our selection in more detail now, and also state more
clearly that a valdiation of further genes would have been desirable. Nevertheless, we have some confidence in our microarry results, too, given the quite strong agreement between microarray and qPCR for the validated genes and considering qPCR results from previous studies that are in line with our microarray data.
See lines 147 to 157 and 370 to 371
4. As the authors mentioned in Discussion section, visceral adipose tissue is more relevant to the physiology of weight loss than
subcutaneous adipose tissue. The authors should describe “subcutaneous adipose tissue” in their title.
We have added “subcutaneous adipose tissue” to our title.
See title
Reviewer 3 Report
The current manuscript by Schübel et al. tried to elucidate the effect of caloric restriction (CR) on abdominal subcutaneous tissue. The study is continuation of their previous study with HALENA trial to study the effect of intermittent caloric restriction with continuous caloric restriction (PMID:27687742). The area of research is indeed interesting and have potential to enrich current understanding of caloric restriction on adipose metabolism.
The manuscript is well written, methods are well described, results are clearly presented. However, the manuscript has some flaws as following
1. The manuscript is mainly descriptive in nature and demonstrated no intension of providing mechanistic insight.
2. Inclusion of healthy donor group as control and comparing them with weight loss groups can provide more useful insights.
3. Serum levels of SFRP2, HILPDA does not match with mRNA expression data diminishing their utility as biomarkers, as collecting adipose biopsies using invasive procedure can be difficult and expensive.
4. Although SFRP2 is a known modulator of WNT-signaling, but there is a clear need of further supportive data to conclude on WNT-signaling. Like in vitro studies using adipocytes and CRISPR-CAS based genetic manipulation of SFRP2 and/or HILPDA could provide valuable insights towards their role in WNT-signaling.
5. Currently, the conclusions are over ambitious. The authors should tone down the conclusion about the modulation of pathways (line 351-352) as most of it is supported by only mRNA expression data.
Author's Reply to Reviewer 3:
Reviewer 3
The current manuscript by Schübel et al. tried to elucidate the effect of caloric restriction (CR) on abdominal subcutaneous tissue. The study is continuation of their previous study with HALENA trial to study the effect of intermittent caloric restriction with continuous caloric restriction (PMID:27687742). The area of research is indeed interesting and have potential to enrich current understanding of caloric restriction on adipose metabolism. The manuscript is well written, methods are well described, results are
clearly presented. However, the manuscript has some flaws as following
1. The manuscript is mainly descriptive in nature and demonstrated
no intension of providing mechanistic insight.
We believe that evidence-based medicine definitvely requires human intervention trials, and that such trials provide mechanistic
insights to some degree. Nevertheless, we do ackowledge that further laboratory-based mechanistic analyses, which were beyond the scope of the present study, are clearly needed. Thus, we have added a further statement on the limitations inherent to
the design of our study to the discussion.
See lines 369 to 371
2. Inclusion of healthy donor group as control and comparing them
with weight loss groups can provide more useful insights.
We agree with the Reviewer that a comparison with lean individuals would have introduced more contrast into our analyses. However, we believe that effects that are realistic in real life situations may be as or even more meaningful than between-group differences from models with extreme group comparisons. We do think that intervention designs, in which intra-indiviudal effects can be detected, have clear advantages compared to mere between-group comparisons with respect to potential confounding. In this regard, we have added a piece of evidence to a field, in which more radical models had already been published. In this regard, we found it particularly interesting, that our study reveals several genes, which have been detected in bariatric surgery studies before. From our perspective, this is a perfect example of consistency, a key criterion for causal assocations in evidence-based nutrition research. Nevertheless, it seems very likely that effects would have been more pronounced with more variation in weight development over time. Thus, we state now in the limitations section that we cannot generalize our results to other settings (e.g. morbidly obese vs. lean persons).
See lines 364 to 366
3. Serum levels of SFRP2, HILPDA does not match with mRNA expression data diminishing their utility as biomarkers, as collecting adipose biopsies using invasive procedure can be difficult and expensive.
This is an important point, and we completely agree with the reviewer. As the reviewer can imagine, we were quite disappointed about the results on circulating HILPDA and SFRP2 ourselvels, as we failed to identify more easily accesible circulating markers derived from our gene expression experiment. There a few examples of succesfull biomarker development from gene expression data, although no fat-tissue derived circulating marker is really used in clinical routine to our knowledge. We have added a critical comment on this well-taken point to our discussion.
See lines 366 to 368
4. Although SFRP2 is a known modulator of WNT-signaling, but there is a clear need of further supportive data to conclude on WNT-signaling. Like in vitro studies using adipocytes and CRISPR-CAS based genetic manipulation of SFRP2 and/or HILPDA could provide valuable insights towards their role in WNT-signaling.
As stated above, we generally agree with the reviewer, but cannot provide functional data here. Such data are needed, but were beyond our possibilities. We can only provide one piece of evidence based on post hoc analyses from a very expensive and comprehenisve human intervention study, and future experimental studies from groups specialized in wet-lab experiments are required. Our focus is on the epidemiological part. In turn, many basic science projects do not provide (human) intervention data. We appreciate that interdisciplinary research covering more experimental approaches would be ideal, but such research requires funding and ressources that we (and many other colleagues) do not have at hand. We have made this clearer in the limitations section.
See lines 369 to 371
5. Currently, the conclusions are over ambitious. The authors should tone down the conclusion about the modulation of
pathways (line 351-352) as most of it is supported by only mRNA expression data.
We agree with this comment (see point 3) and have toned down the conclusion. We have also shortened our final statement on
potential pathways in the abstract.
See lines 375 to 379